# Exploring the spatial association between the distribution of temperature and urban morphology with green view index

Ta-Chien Chan[1,2,3,4]*, Ping-Hsien Lee[1], Yu-Ting Lee[1], Jia-Hong Tang[1]

1 Research Center for Humanities and Social Sciences, Academia Sinica, Taipei, Taiwan, 2 Institute of Public Health, School of Medicine, National Yang Ming Chiao Tung University, Taipei, Taiwan, 3 Department of Public Health, College of Public Health, China Medical University, Taichung Campus, Taiwan, 4 School of Medicine, College of Medicine, National Sun Yat-sen University, Kaohsiung, Taiwan

* tachien@sinica.edu.tw

**Data Availability Statement:** The data and code used in this study have been uploaded to a public depository (https://doi.org/10.6084/m9.figshare.

## Abstract

Urban heat islands will occur if city neighborhoods contain insufficient green spaces to create a comfortable environment, and residents' health will be adversely affected. Current satellite imagery can only effectively identify large-scale green spaces and cannot capture street trees or potted plants within three-dimensional building spaces. In this study, we used a deep convolutional neural network semantic segmentation model on Google Street View to extract environmental features at the neighborhood level in Taipei City, Taiwan, including the green vegetation index (GVI), building view factor, and sky view factor. Monthly temperature data from 2018 to 2021 with a 0.01˚ spatial resolution were used. We applied a linear mixed-effects model and geographically weighted regression to explore the association between pedestrian-level green spaces and ambient temperature, controlling for seasons, land use information, and traffic volume. Their results indicated that a higher GVI was significantly associated with lower ambient temperatures and temperature differences. Locations with higher traffic flows or specific land uses, such as religious or governmental, are associated with higher ambient temperatures. In conclusion, the GVI from street-view imagery at the community level can improve the understanding of urban green spaces and evaluate their effects in association with other social and environmental indicators.

## Introduction

Population agglomeration in metropolitan cities is a worldwide trend and drives socioeconomic development [1]. Asian cities like Hong Kong [2] and Taipei [3] have a compact layout, high urban population densities, and overcrowded buildings. Green spaces or green planting measures act as regulators, helping to mitigate the urban heat island effect [3, 4]. Furthermore, a lack of green spaces affects people's well-being and mental health and can even increase the risk of mortality [5, 6]. Satellite image-derived metrics [5] such as the Normalized Difference Vegetation Index (NDVI) and Enhanced Vegetation Index (EVI), which quantify greenness at the grid level, and geographic information systems, used to calculate the spatial accessibility of

24922275; https://doi.org/10.6084/m9.figshare.
24922278).

**Funding:** This study was supported by a grant
from the National Science and Technology Council
of Taiwan (grant number MOST 111-2121-M-001-
002). The funder played no role in the study
design, data collection and analysis, decision to
publish, or manuscript preparation.

**Competing interests:** The authors have declared
that no competing interests exist.

park green spaces at the community level [7] are popular ways of computing proxy indicators of urban green space. Satellite imagery offers continuous spatial coverage of greening data, highly correlated with the spatial accessibility of parks or other open green spaces in a community. However, satellite images must be taken from above, resulting in limitations in spatial resolution and shooting angles. Tight building layouts may obscure ground-level images and vertical views of buildings. Bird's-eye-view satellite data cannot effectively capture street trees, shrubs, and plants on roads, green curtains, and green façades on the vertical sides of buildings. In cities, in addition to large parks and green spaces, street trees or plants cultivated by residents are closer to their daily living environments and may directly contribute to their thermal comfort, namely, the subjective evaluation of the thermal environment, noise relief, and air purification.

To improve the measurement of city green spaces, Li et al. [8] proposed a new method that adopts an image segmentation approach to compute the green-view index (GVI) from panorama images from Google Street View (GSV) in 2015. The evolution of the GVI from GVI 1.0 (https://github.com/mittrees/Treepedia_Public) to GVI 2.0 (https://github.com/billcai/treepedia_dl_public) shows a marked improvement in identifying green vegetation and reducing the false recognition of green background. Previous research [9] has shown that, compared with NDVI, the correlation between GVI and NDVI is relatively low, mainly due to the limitations of the spatial resolution of satellite imagery and the essential limitations of GSV, which needs to be collected along the road. In this study, we increased the GSV sampling rate to provide finer measures of GVI from a pedestrian perspective.

Urban green spaces can mitigate urban heat island (UHI) effects and play an important role in cooling urban spaces [9]. Unlike previous studies that focused on urban parks or satellite imagery, this study incorporated new ecological indicators, including GVI, Building View Factor (BVF), and Sky View Factor (SVF), calculated from GSV and other environmental data, including satellite images, altitude, population, building areas, traffic volume, and land-use information. These three indicators are closely related to factors such as regional temperature regulation, socioeconomic population distribution [10], air pollution [11, 12], and housing prices [13]. According to an Australian study [14] that used Google Street View to extract SVF, there are differences between urban and rural SVF. Sky View Factor and shade were highly correlated with population traits and the heat vulnerability index (HVI). A study in Los Angeles, U.S.A., reported that socioeconomic status in the community was highly correlated with the green index [15]. These GSV-derived indicators can be used as urban morphological indicators at a fine community scale.

In a densely populated city such as Taipei, it is difficult to find large areas of green space because of the building density. The NDVI has limited efficacy in quantifying greening effects at the community level in urban areas. Anthropogenic factors contributing to increased ambient temperatures, such as different land uses and traffic volumes, influence urban temperatures. In this study, we innovated by incorporating three GSV-derived indicators—land-use data, population density, altitude data, satellite imagery data, and traffic volume data from vehicle detectors—to explore the association between green-view indexes and ambient temperature at the community scale.

## Methods

### Ethics approval and consent to participate

This study was conducted using non-human subjects. Therefore, the requirement for ethical approval was waived.

**Google Street View (GSV) data collection.** We first sampled GSV points at intervals of 30 m from the commercial road network data in Taipei City, Taiwan. We then applied the Google Map API with an authentication code to retrieve our sample points' latest panorama images from 2018 to 2022 (https://developers.google.com/maps/documentation/streetview/overview).

We split the panoramic image into six separate images for each sample point, with pitch angles of 0˚ and 45˚. Thus, 12 images were obtained for further GVI computation at each point. The resolution of each image is 224 × 224 pixels. A total of 86,637 sample points and 1,039,644 images were used in this study.

**Definition and computation of GVI 2.0.** The GVI is one way to measure green vegetation from pedestrians' horizontal and vertical perspectives along roads. In this study, we applied a deep convolutional neural network (DCNN) semantic segmentation model from Treepedia 2.0 [16] to compute the GVI 2.0, which differentiated it from the first-generation GVI proposed in 2015 [8]. The main difference between GVI 1.0 and GV 2.0 is that the GVI 2.0 algorithm improves the accuracy of tree cover identification by learning from many publicly available labeled street images. In the image segmentation, the pixels were classified as green or non-green. At each sampling point, a total of six azimuths (0˚, 60˚, 120˚, 180˚, 240˚, and 300˚) and two elevations (vertical) angles (0˚ and 45˚) of GSV images were measured, and a total of 12 photo sets were collected. The formula for computing GVI 2.0 was:

$$Green\ View\ Index\ (GVI\ 2.0) = \frac{\sum_{i=1}^{12} pixels_{g_i}}{\sum_{i=1}^{12} pixels_{t_i}} \times 100(\%)$$

Where $pixels_{g_i}$ is the number of green vegetation pixels in one image, and $pixels_{t_i}$ is the total pixel number in one image.

**Computation of sky view factor (SVF) and building view factor (BVF).** The SVF and BVF, representing the view from the ground, were computed from fisheye images of the GSV. Fisheye images were processed using SegNet for image segmentation [17], and pixels of the sky and buildings were identified. The sum of the tree view factor (TVF), SVF, and BVF was 100%. TVF and GVI were highly correlated. Therefore, we did not consider the TVF in our model. The SVF is a dimensionless measure that varies between 0 and 1 and is used to characterize sky openness in fisheye images [18]. A value of 1 represents the full sky without any blocks, and 0 indicates that buildings or trees completely block the image and the sky cannot be seen.

$$Sky\ View\ Factor\ (SVF) = \frac{\sum_{i=0}^{n} \omega \times sky(i)}{\sum_{i=0}^{n} \omega} \times 100(\%)$$

where n is the total number of pixels, $\omega$ is the weight associated with each pixel, and sky(i) is a function determining whether this pixel is the sky.

The measure (BVF) represents the compactness of buildings along the road and ranges from 0 to 1. Higher values represent a higher percentage of buildings in the fisheye images.

$$Building\ View\ Factor\ (BVF) = \frac{\sum_{i=0}^{n} \omega \times building(i)}{\sum_{i=0}^{n} \omega} \times 100(\%)$$

where n is the total number of pixels, $\omega$ is a weight associated with each pixel, and building(i) is a function that determines whether this pixel shows buildings.

**Normalized Difference Vegetation Index (NDVI).** The NDVI is an indicator commonly used to quantify the degree of greenness in satellite images. The advantage of the NDVI is that it can help identify greenness over a large area, which can benefit urban planning. In this study, we applied the Google Earth Engine (GEE) [19] to collect two types of satellite images as alternative greenness indicators: the Moderate Resolution Imaging Spectroradiometer (MODIS) and Landsat 8. The spatial resolution of MODIS is 250 m, and that of Landsat 8 is 30 m. The NDVI ranges from -1 to 1.

$$Normalized\ Difference\ Vegetation\ Index\ (NDVI) = \frac{NIR - RED}{NIR + RED}$$

Here, RED represents the spectral reflectance acquired in the red (visible), and near-infrared (NIR) stands for the spectral regions.

We matched the GSV year and month information of the sampling points with the year and month information corresponding to the two spatial resolutions of the NDVI from 2018 to 2022. Then, we compared the correlation between the GSV-derived GVI and NDVI in Taipei City.

**The grid-based temperature data.** The original temporal resolution of the temperature data at a spatial resolution of 0.01˚ is from the Taiwan Climate Change Projection Information and Adaptation Knowledge Platform (TCCIP, https://tccip.ncdr.nat.gov.tw/index_eng.aspx) from 2018 to 2021. The temperature data used in this study included the monthly average temperature and the monthly temperature difference computed from the daily difference between the maximum and minimum temperatures. As shown in Fig 1, there were 232 grids in the study area.

**Land-use data.** Land-use survey data for 2021 were obtained from the National Land Surveying and Mapping Center of the Ministry of the Interior, Taiwan (https://www.nlsc.gov.tw/en/). The classification of land-use data included nine classes in the first tier, 48 classes in the second tier, and 93 classes in the third tier. Official land-use classifications are based on the law and previous land-use surveys. Each upper-level classification has different subcategories. The first-tier category is broad and includes agriculture, forestry, transportation, water conservation, construction, public utilities, entertainment, and mineral salt utilization. However, because the study site is located in the metropolitan area of Taipei City in northern Taiwan, we selected the 19 land-use types with the highest land-use area ratio for statistical analysis. The types of land use include farms, forests, mass rapid transit (MRT), roads, rivers, water storage facilities, commercial areas, residential areas, mixed commercial residential areas, industrial areas, religious sites, funeral facilities, parks, leisure facilities, vacant land, government agencies, schools, social welfare facilities, and public facilities. We computed each TCCIP grid's percentage of each land-use type using QGIS 2.28 [20].

**Human activity.** This study uses two types of data to represent human activities in communities. The first was the hourly traffic volume data from 737 automated vehicle detection stations in Taipei City from the Taipei City Traffic Engineering Office from 2018 to 2021 (Fig 1). The different road levels are shown in Fig 1. To ensure consistency in spatiotemporal resolution, we calculated the monthly averages of the hourly traffic volume in each grid. The second dataset comprises static population data from June 2022 from the socioeconomic database maintained by the Ministry of the Interior of Taiwan. At that time, Taipei City had approximately 2.5 million registered residents. The spatial resolution of the population data was within the basic statistical area (BSA). The QGIS spatial analysis function was used to compute the number of populations in the grids.

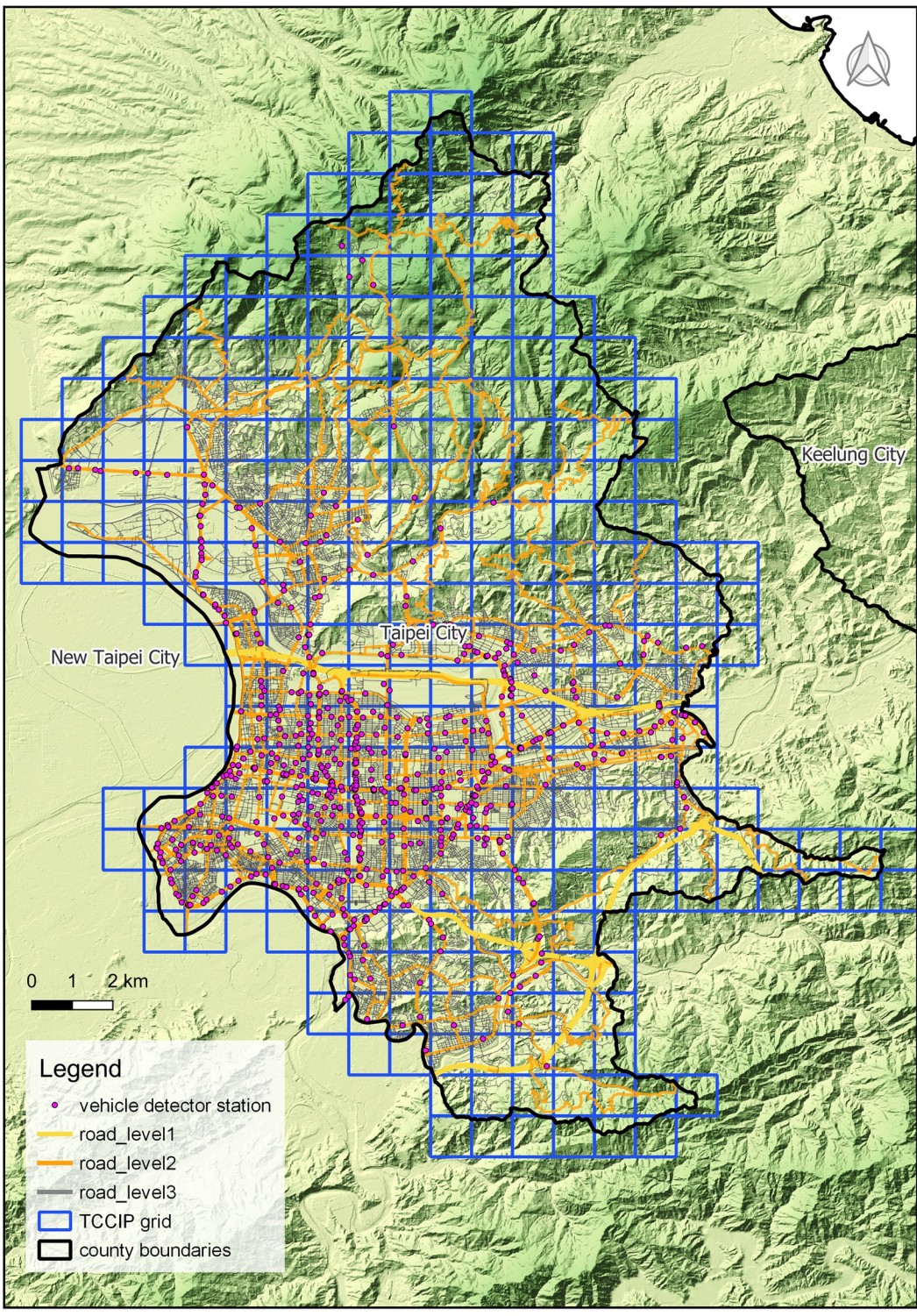

**Fig 1. Spatial distribution of vehicle detectors, road networks, and grids with temperature data from the Taiwan Climate Change Projection Information and Adaptation Knowledge Platform (TCCIP).** Layers such as vehicle detectors (https://data.gov.tw/dataset/135705), road networks (https://data.gov.tw/dataset/156810), county boundaries (https://data.gov.tw/dataset/7442), and 20m Digital Terrain Model (DTM, https://data.gov.tw/dataset/103884) are all from the Taiwan Open Data Platform (https://data.gov.tw/) which is followed by Open Government Data License, version 1.0 (https://data.gov.tw/license). The License is compatible with the Creative Commons Attribution License 4.0 International.

**Altitude and size of the building area.** As altitude is related to temperature, we used open data from the digital terrain model at a resolution of 20 m in 2019 from the Ministry of Interior, Taiwan (https://data.gov.tw/dataset/103884). The zonal statistical function in QGIS was used to compute the average altitude of the grids. In addition, we computed the building areas within each grid by using the one-thousandth numerical topographic map of Taipei City from the Taipei City Department of Urban Development in 2017.

**Statistical model.** In this study, we converted all explanatory data into grid units to match the TCCIP grid of the dependent variables of interest, including the monthly average temperature and monthly average temperature difference. The explanatory variables included the season, GSV-derived indicators (GVI, SVF, and BVF), NDVI from satellite imagery (MODIS and Landsat 8), land use, human activity, altitude, and building area. We used two statistical models to analyze the factors influencing urban temperature: a linear mixed model (LMM) and a geographically weighted regression (GWR). For LMM, we applied R software v. 4.2.0 [21] with the "lme4" package [22] to consider the repeated measurement of temperature data across different seasons and treat the random effect for the grids. Due to the collinearity issue between GVI and NDVI in the same model, we treated GVI and the two types of NDVI with the same covariates in the same model. We used the smallest Akaike information criterion (AIC) to select the model. In both the LMM and GWR models, the green space indicator using GVI had the lowest AIC, and we included the GVI in our final model. Considering the GVI, we treated the seasons as categorical variables to account for their effect on the average temperature or the temperature difference.

$$Y_i = \beta_0 + \sum_{k=1}^{m} \beta_k X_{ik} + b_i + \varepsilon_i, i = 1, 2, \ldots, 232$$

where $Y_i$ is the monthly average temperature or the monthly average temperature difference in grid $i$. $X_{ik}$ was the value of the $k$th explanatory variable in grid $i$ and $m$ is the number of explanatory variables. $\beta_0$ was the intercept term. $X_k$ included seasons, green metrics (GVI, NDVI (MODIS), or NDVI(Landsat 8)), SVF, proportions of land use types (19 types), population, traffic volume, altitude, and building area. The $\beta_k$ was the regression coefficient for the $k$th explanatory variable. $b_i$ represented the grid-specific random effect that is independently normally distributed with a zero mean and variance $\sigma_b^2$. The $\varepsilon_i$ was the random error term that is independently normally distributed with a zero mean and a common variance of $\sigma^2$.

In addition, various factors may have different degrees of influence on the average temperature or the temperature difference in a specific local grid. Thus, we considered spatial heterogeneity using GWR to show the different estimations of the explanatory variables for each grid. We ran the analysis using the R package "GWmodel" [25].

$$Y_i = \beta_0(u_i, v_i) + \sum_{k=1}^{m} \beta_k(u_i, v_i) X_{ik} + \varepsilon$$

where $(u_i, v_i)$ denoted the coordinates of grid $i$, $\beta_0(u_i, v_i)$ represented the intercept value, and $\beta_k(u_i, v_i)$ is a set of values of parameters at grid $i$. This model allowed the parameter estimates to vary across spaces and was likely to capture local effects. In each of the local regression equations, we used a Gaussian weighting scheme to assign a weight of one to each target grid. As the distance from the regression feature increased, the weights for the surrounding grids smoothly and gradually decreased.

## Results

An overview of the green view index in Taipei City from 2018 to 2022 is shown in Fig 2A. Taipei City lies in the Taipei Basin and the mountainous areas on the eastern side provide higher

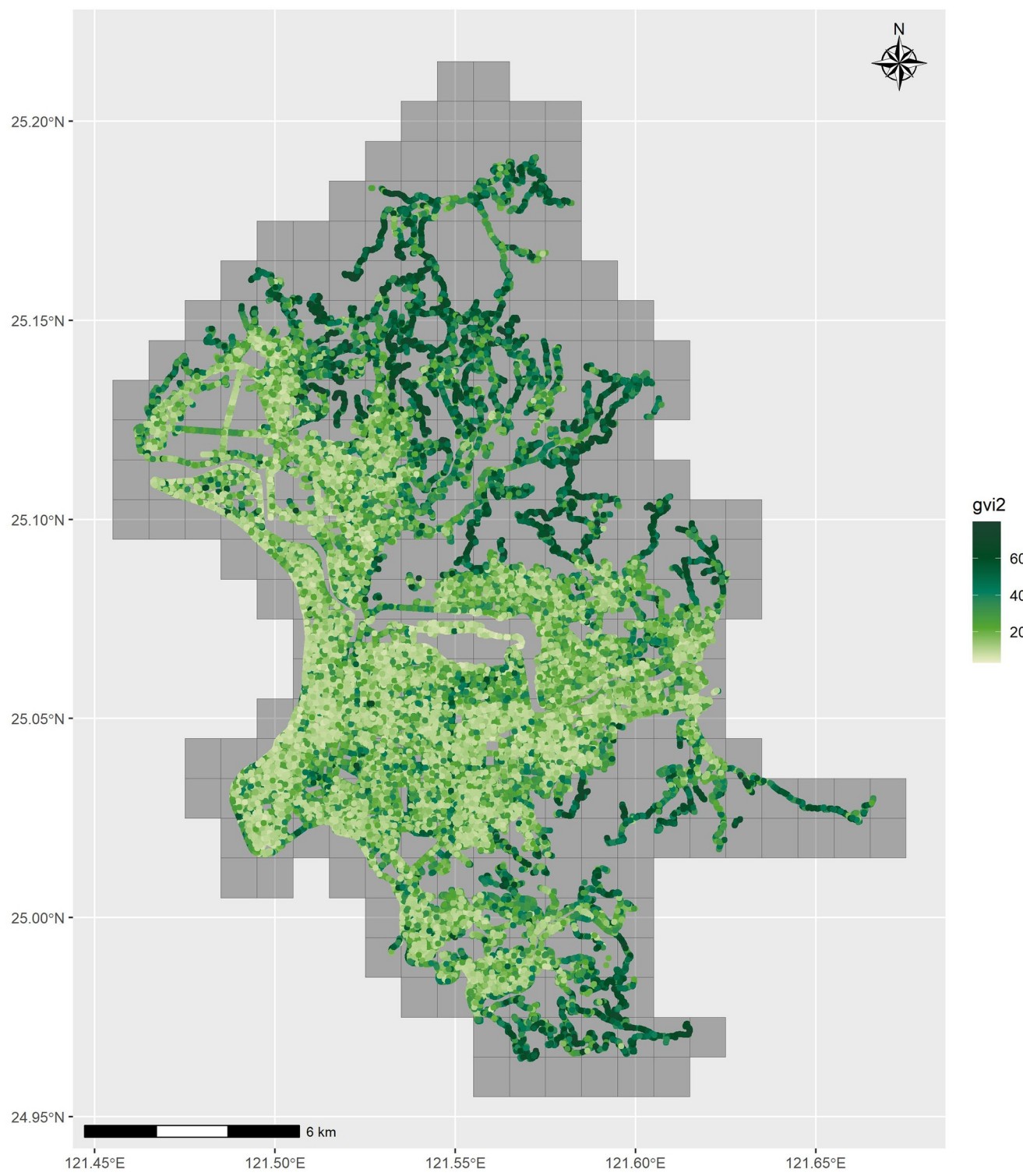

**Fig 2. Overview of the green view index in Taipei City from 2018 to 2022.** (A) Green view index estimated during the study period, at 30 m intervals. (B) Green view Index data distribution (mean: 30.17%).

GVI. Within the densely populated city area, the GVI has spatial heterogeneity. The mean GVI during the study period was 30.17% (Fig 2B). Due to the filming time of the GSV, we obtained the latest GSVs for computing the GVI from different seasons in different spaces. Stratification of the GVI into four seasons can help us understand the distribution of the green view (Fig 3). Although green vegetation may shed its leaves in the fall or winter, the GVI remains relatively stable, particularly in subtropical cities such as Taipei. This is because, even though the leaf pixels are lost, image segmentation still identifies pixels belonging to other parts of the vegetation, such as the trunk and branches. Fig 4 shows the average monthly temperatures during the four seasons from 2018 to 2021. The spatial trend shows that northern Taipei City is much cooler than the rest of the city area, and the center of Taipei City is hot during all four seasons.

Table 1 provides descriptive statistics of the variables included in the models. The two dependent variables were the average monthly temperature (21.51 ± 2.94˚C) and the average monthly temperature difference (5.69 ± 0.98˚C). The mean and standard deviation of green view index 2, sky view factor, and building view factor are 30.17 ± 14.36%, 42.66 ± 11.49%, and 22.74 ± 19.83%. The mean NDVI values obtained from MODIS and Landsat 8 are 0.51 and 0.2. The average temperatures in winter, spring, summer, and fall were 17.4, 22.15, 27.70, and 22.74˚C, respectively. The average altitude is 181.35 m. The average population in the grid was 10,192, and the average building area was 147,928.32 m$^2$. The average hourly traffic volume was 747.44 vehicles per hour. The top five land-use types in the grids of Taipei City were forests (36.8%), roads (12.35%), residential areas (8.85%), agricultural fields (7.39%), and mixed commercial and residential areas (4.75%).

The spatial resolution affects the green view. This study compared the correlations between the GVI 2 and two other NDVI values at the grid level (Table 2). The overall correlation was higher between GVI 2 and Landsat 8 (r = 0.756, p<0.001) than between GVI 2 and MODIS (r = 0.577, p<0.001). The correlation between the Landsat 8 and MODIS data was 0.577 (p<0.001). The correlations were similar for spring, summer, and fall but slightly weaker in winter.

In the linear mixed model (Table 3), we controlled for major seasonal effects on the average temperature. The coefficient of GVI 2 was -0.049 (p<0.001), indicating that a higher green view index was negatively associated with the average temperature. We observed that religion-related land use was positively associated (coefficient = 0.228, p = 0.034) with average temperature. In addition, grids with higher traffic volumes (coefficient: 0.001, p<0.001) were positively associated with average temperature. In Table 4, we applied the GWR with the same variables to evaluate the association. The direction of the median estimations from GWR was like that of the LMM. The median values of GVI 2, religion-related land use, government land use, and traffic volume were -0.016, 0.008, 0.004, and 0.00009, respectively. The coefficients in the last column were obtained using simple linear regression. Government land use was significant in the simple linear regression and positively correlated with average temperature. The overall model performance in terms of the adjusted R-squared was 96.6% for the LMM and 95.8% for the GWR. The two significant predictors selected from the GWR are shown in Fig 5. The estimation of GVI 2 in Fig 5A for northern and southeastern Taipei City showed that a higher GVI was associated with a lower average temperature. The traffic volume estimation is shown in Fig 5B, and a few of the significant grids overlap with those in Fig 5A, indicating that a higher traffic volume is associated with a higher average temperature.

In the Appendix, we describe the results related to the temperature differences. As shown in S1 Appendix, the average monthly temperature difference was high in the spring and summer and the south of Taipei City. The results of the LMM (S2 Appendix) on the temperature difference showed that the difference decreased with each year (coefficient: -0.099, p<0.001). Higher GVI2 and SVF values were negatively associated with the temperature differences.

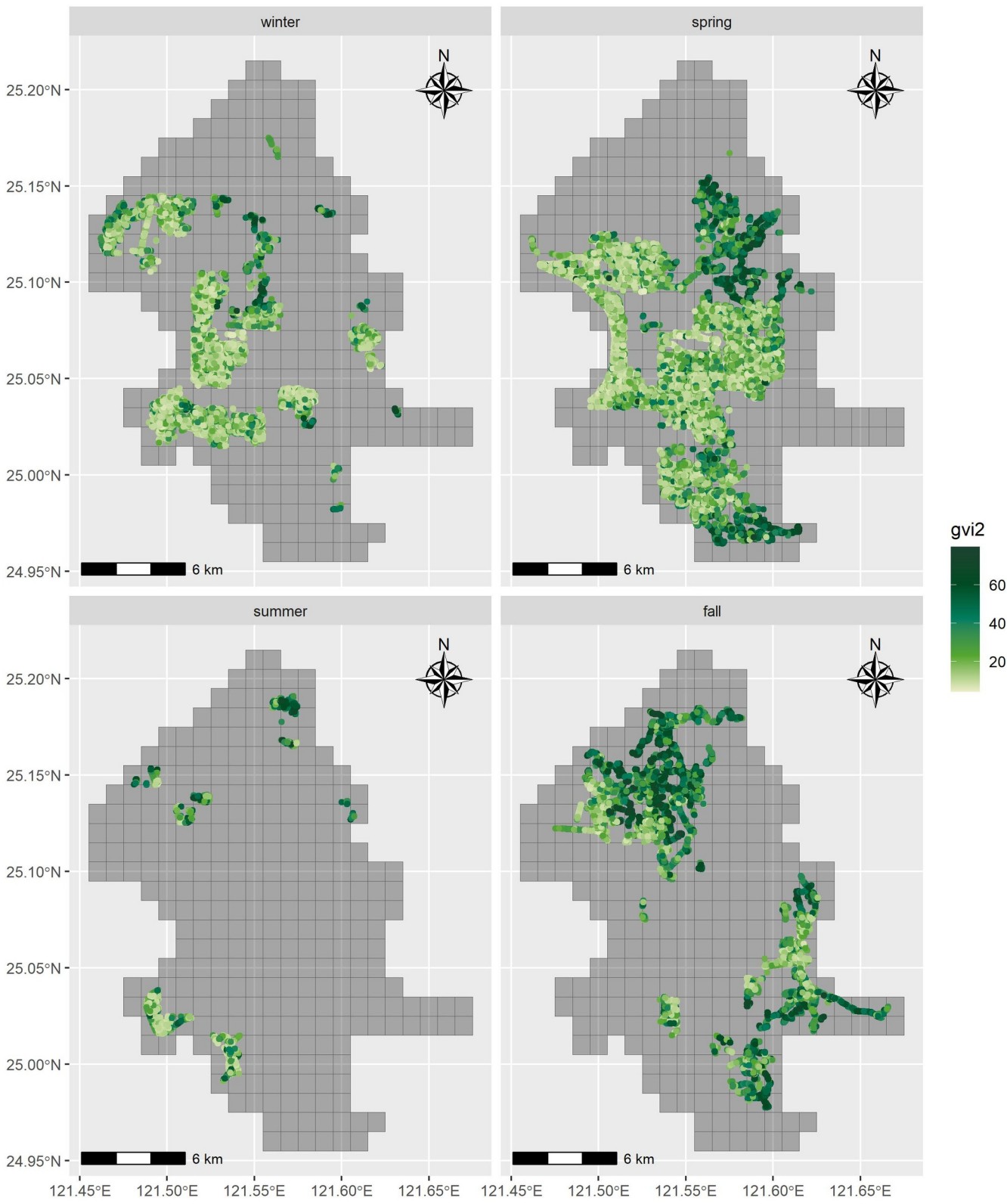

**Fig 3. The stratified visualization of the green view index by four seasons.**

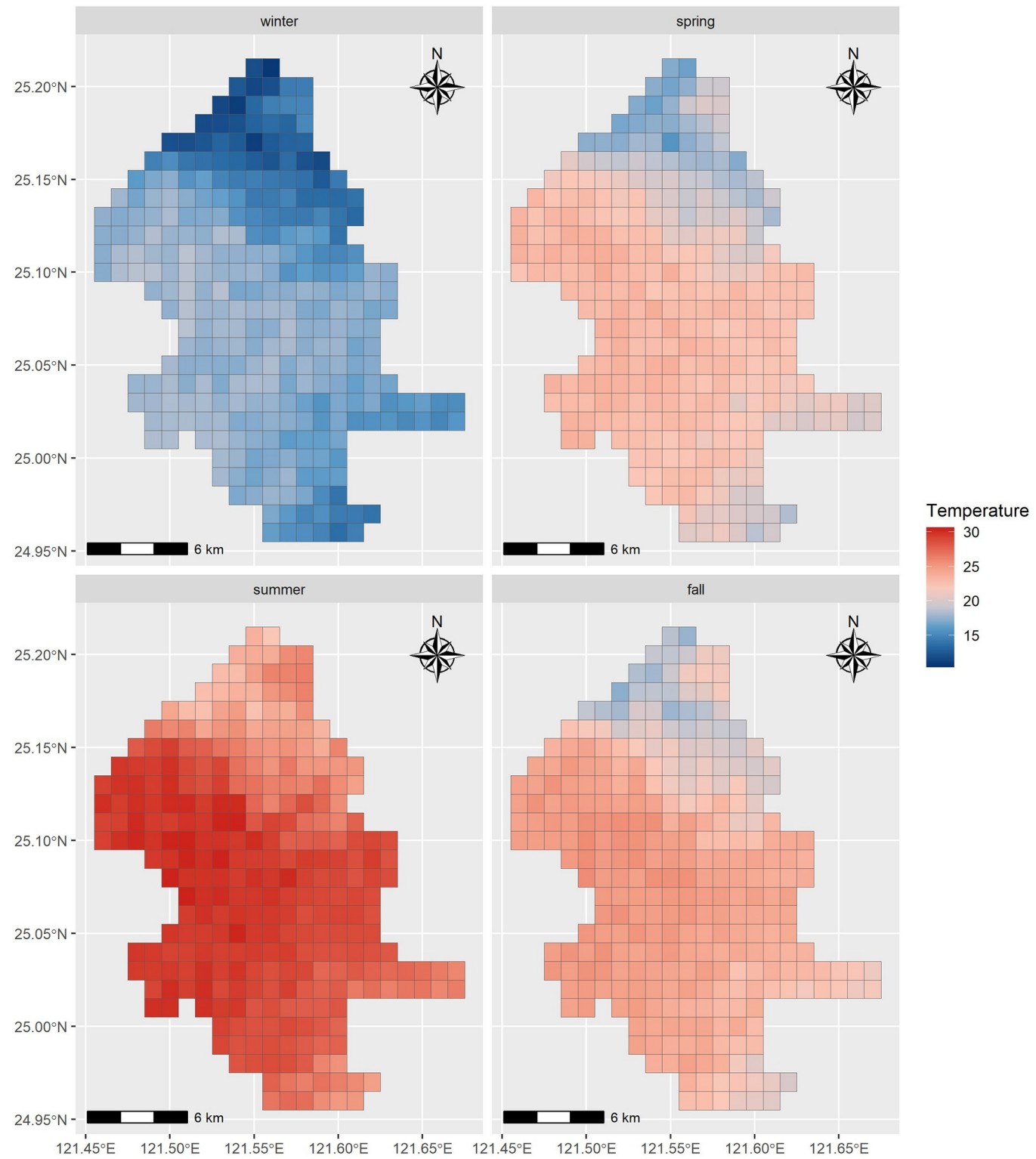

**Fig 4. Average monthly temperature in four seasons from 2018 to 2021.**

**Table 1. Descriptive statistics of the variables used in the models.**

| Variables | Mean ± standard deviation |
|---|---|
| Average monthly temperature (˚C) | 21.51 ± 2.94 |
| Average monthly temperature difference (˚C) | 5.69 ± 0.98 |
| Green view index 2 (%) | 30.17 ± 14.36 |
| Sky view factor (%) | 42.66 ± 11.49 |
| Building view factor (%) | 22.74 ± 19.83 |
| NDVI MODIS (250m) | 0.51 ± 0.22 |
| NDVI Landsat 8 (30m) | 0.2 ± 0.1 |
| Season | |
| Winter (˚C) | 17.40 ± 1.19 |
| Spring (˚C) | 22.15 ± 1.39 |
| Summer (˚C) | 27.70 ± 2.11 |
| Fall (˚C) | 22.74 ± 1.94 |
| Altitude (meters) | 181.35 ± 244.09 |
| Population (persons) | 10192 ± 12990 |
| Building area (square meters) | 147928.32 ± 148890 |
| Traffic volume (vehicles/hr) | 747.44 ± 359.60 |
| Land use within TCCIP grids | |
| Agriculture field (%) | 7.39 ± 11.56 |
| Forest (%) | 36.8 ± 32.58 |
| Mass rapid transit (MRT) (%) | 0.37 ± 1.25 |
| Road (%) | 12.35 ± 10.96 |
| River (%) | 2.87 ± 6.64 |
| Water storage facilities (%) | 0.14 ± 0.58 |
| Commercial area (%) | 3.18 ± 4.98 |
| Residential area (%) | 8.85 ± 8.39 |
| Mixed commercial residential area (%) | 4.75 ± 7.12 |
| Industrial area (%) | 0.27 ± 0.87 |
| Religious sites (%) | 0.46 ± 0.62 |
| Funeral facilities (%) | 1.08 ± 3.92 |
| Park (%) | 3.96 ± 5.59 |
| Leisure facilities (%) | 0.65 ± 2.73 |
| Vacant land (%) | 1.23 ± 2.35 |
| Government agencies (%) | 2.04 ± 6.2 |
| School (%) | 3.98 ± 7.1 |
| Social welfare facilities (%) | 0.09 ± 0.21 |
| Public facilities (%) | 0.21 ± 0.49 |

Land use for funerals was positively associated with temperature differences. The results of the simple linear regression and the GWR (S3 Appendix) are consistent with those of the LMM. The adjusted R-squared values for LMM and GWR for temperature differences were 71.8% and 75.7%, respectively. In the results of the GWR estimation in space (S4 Appendix), a higher GVI 2 was negatively associated with the temperature difference in northern Taipei and positively associated with the temperature difference in southern Taipei. Higher SVF showed only negative associations in a few grids in northern Taipei. Moreover, a higher percentage of funeral land use was positively associated with temperature differences in northern Taipei.

**Table 2. The correlations among green view index 2.0, NDVI from Landsat 8 satellite imagery, and NDVI from MODIS satellite imagery.**

| Overall correlation | GVI2 | NDVI Landsat 8 | NDVI MODIS |
|---|---|---|---|
| GVI2 | - | 0.756*** | 0.577*** |
| NDVI Landsat 8 | 0.756*** | - | 0.686*** |
| NDVI MODIS | 0.577*** | 0.686*** | - |
| In winter | | | |
| GVI2 | - | 0.567*** | 0.480*** |
| NDVI Landsat 8 | 0.567*** | - | 0.553*** |
| NDVI MODIS | 0.480*** | 0.553*** | - |
| In spring | | | |
| GVI2 | - | 0.759*** | 0.539*** |
| NDVI Landsat 8 | 0.759*** | - | 0.690*** |
| NDVI MODIS | 0.539*** | 0.690*** | - |
| In summer | | | |
| GVI2 | - | 0.763*** | 0.532*** |
| NDVI Landsat 8 | 0.763*** | - | 0.728*** |
| NDVI MODIS | 0.532*** | 0.728*** | - |
| In fall | | | |
| GVI2 | - | 0.786*** | 0.586*** |
| NDVI Landsat 8 | 0.786*** | - | 0.628*** |
| NDVI MODIS | 0.586*** | 0.628*** | - |

***$p < 0.001$

## Discussion

Urban greening is an important strategy for reducing urban ambient temperatures and can play a part in reducing noise and improving air quality. Unlike previous approaches that used land surface temperature and greenness data from satellite imagery, we innovatively leveraged grid-based temperature data from long-term meteorological ground observations, GSV-derived green-view indicators, and seasonal and land-use information. By densely sampling GSVs from Taipei City's dense road network, we constructed a GVI distribution from the perspective of pedestrians. In our results, the GVI was negatively correlated with mean air

**Table 3. Results of the linear mixed model on average monthly temperature.**

| Variables | Estimate | Std. Error | P-value | VIF |
|---|---|---|---|---|
| GVI2 | -0.049 | 0.007 | <0.001 | 1.143 |
| Season (ref = fall) | | | | 1.033 |
| Spring | -1.782 | 0.101 | <0.001 | |
| Summer | 4.849 | 0.191 | <0.001 | |
| Winter | -6.997 | 0.106 | <0.001 | |
| Land use: religion | 0.228 | 0.106 | 0.034 | 1.015 |
| Land use: government | 0.009 | 0.009 | 0.304 | 1.007 |
| Traffic volume | 0.001 | 0.0001 | <0.001 | 1.061 |

Adjusted R-squared: 96.6%; AIC: 1180.8

**Table 4. Results of the geographically weighted regression and simple linear regression on average monthly temperature.**

| Variables | Min. | Q1 | Median | Q3 | Max. | Coefficients[a] |
|---|---|---|---|---|---|---|
| GVI2 | -0.089 | -0.025 | -0.016 | -0.009 | 0.007 | -0.055** |
| Season (ref = fall) | | | | | | |
| Spring | -2.232 | -1.823 | -1.702 | -1.534 | -0.789 | -1.45** |
| Summer | 4.174 | 4.646 | 4.782 | 5.092 | 5.639 | 5.24** |
| Winter | -7.608 | -7.043 | -6.95 | -6.712 | -6.282 | -6.613** |
| Land use: religion | -0.13 | 0.001 | 0.008 | 0.021 | 0.143 | 0.222** |
| Land use: government | -0.312 | -0.05 | 0.004 | 0.046 | 0.806 | 0.009* |
| Traffic volume | -0.00033 | -0.00002 | 0.00009 | 0.00029 | 0.00075 | 0.00056** |

Adjusted R-squared: 95.8%; AIC: 1129.096

**p<0.001;

*p<0.05

[a]: estimated from simple linear regression

[b]: estimated from geographically weighted regression

temperature and temperature difference, which means that green infrastructure in urban areas helps regulate ambient temperature smoothly. Human activities, such as traffic volume, and specific land-use types, such as religion and government, are associated with high temperatures. Using the GWR model, we determined the spatial impacts of different human activities on the local temperatures.

In our study, the associations between GVI and NDVI, both overall and across seasons, ranged from moderate to high according to the spatial resolution of the data. Landsat 8 has a

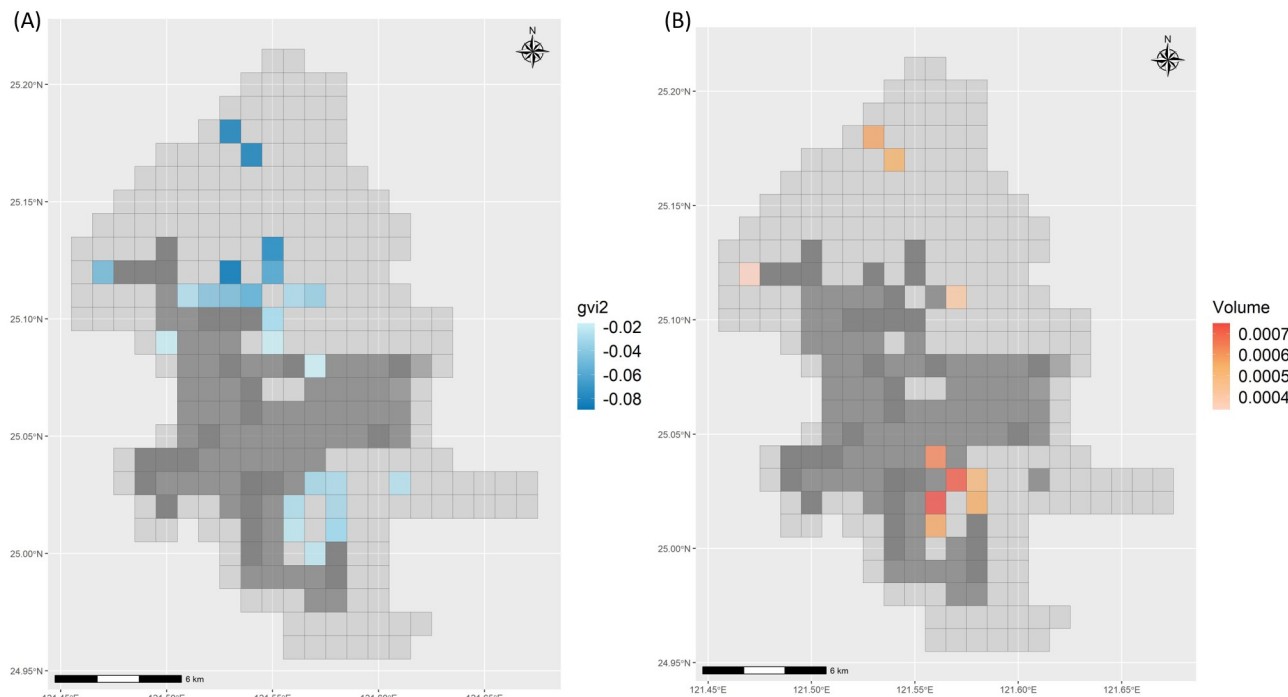

**Fig 5. Geographical distributions of the selected significant predictors of monthly average temperature.** (A) Green view index. (B) Traffic volume.

resolution of 30 m, equal to the GSV sampling interval. Therefore, the correlation was 0.75, except in winter (r = 0.567). The MODIS data have a resolution of 250 m, much coarser than that of the GSV. Thus, the correlation coefficient decreases to 0.577. The correlations between the green-view index from street-view images and the NDVI have varied in different studies. A study conducted in northeast China used Tencent Map Street View images to compute the GVI with a sampling interval of 100 m and compared it to the NDVI of Landsat 5 at a resolution of 30 m. The association with greenness within 1000 m was 0.85 [23]. Another study conducted in Ireland showed a high correlation (r = 0.85) between GVI and NDVI within a 500-m buffer of an air monitoring station [24]. However, other studies reported weak associations. For example, a study conducted in Canada by the Canadian Urban Environmental Health Research Consortium used Landsat 5 and Landsat 8 NDVI and GVI from GSV data [25]. Their association was only 0.14, but GVI (20.1%) was more relevant in explaining air pollution exposure than NDVI (1.4%). The authors concluded that GVI may be a more sensitive indicator of tree exposure. Different urban morphologies may affect the urban greenness measurements from an overhead or pedestrian perspective.

A previous study reported the association between greenness and ambient temperature. As in our Taipei Basin study, the researchers selected only three days of land surface temperature, associated green space coherence, and local climatic zone [3]. They reported that the cooler environments were associated with increased greenery and clustering of green spaces. Our study identified the benefits of cooling from increased GVI across years and seasons. Southwest Taipei City is cooler than other regions of Taipei City because of the highly compact layout of the buildings. The GVI had significant cooling effects, especially in the northern and southeastern regions close to the foot of the mountains. Another study [26] conducted in the West District of Taichung City, Taiwan, used GSV to compute the SVF and GVI from 50 sample locations and applied on-site questionnaires and meteorological measurements. The study observed that the SVF was positively correlated with the physiological equivalent temperature (PET) and thermal sensation vote (TSV) and negatively correlated with street-level perceived shade. The GVI is negatively correlated with PET, indicating that higher greenness levels can result in cooling and greater thermal comfort. Although our study did not measure PET, we observed a correlation between ambient temperature and PET and a similar negative correlation between GVI and temperature. However, in our study, a higher SVF was associated with lower temperature differences, not the mean temperature. Our number of GSVs was 86,637 points, much higher than in the above-mentioned study, with only 50 sampling points. Therefore, we believe that our data can further explain the changes in GSV-derived indicators and the impact of temperature changes in different areas within the city.

Based on previous longitudinal observations, increased built-up areas and urban population density are associated with increased land surface temperature [27]. In this study, we used the latest GSV, which does not reflect temporal changes in land use owing to discontinuous filming dates in different city areas. However, the cross-sectional association between the current GVI and the corresponding average temperature within cities still reflects the spatial gaps in our cooling strategy. The southwestern part of Taipei City has a high population density and is the hottest area. However, the GVI did not have a significant cooling effect here. The GVI in the northern part of the city was inversely associated with mean temperature and temperature differences, while temperature differences were positively correlated with funeral land use. In the southern part of the city, heavy traffic volume was positively correlated with higher ambient temperatures, while temperature differences were positively correlated with the GVI. Most of these significant areas are close to the foot of the mountain, at the junction of urban and rural areas, where the temperature is intermediate. Overall, increased green vegetation in cities had a cooling effect; however, this effect was not as pronounced as that in the surrounding

urban areas. Using new types of data, such as GSV-derived indicators and continuous spatial coverage of grid-based temperature data with other environmental and anthropogenic information, and applying spatially oriented GWR models can help researchers and policymakers quantify local influencing factors on ambient temperature or temperature difference. The level of GVI at the community level can help improve green infrastructure planning and evaluate greening effects.

## Limitations

The data frequency of the GSV was primarily from a street view car and depended on the filming schedule. Thus, unlike satellite imagery, which had a full spatial extent and regular temporal intervals, the temporal and spatial extents of street-view imagery were discontinuous. Therefore, we had to combine our GVI observations with GSV data from 2018 to 2022 to obtain a complete picture of the GVI of the entire city. Although we had a high time resolution for the temperature data, we had to match the GSV in the resolution month and lose the data linkage for those without GSV data.

The GSV imposes a second limitation: the shooting must occur along a road. In large green space areas, GSV can only collect outline data on green vegetation. This may underestimate the cooling effect of green spaces like parks and forests. Our study used NDVI and park and forest land-use data. The NDVI affected the degree of cooling. However, the overall model performance was no better than that of the GVI model. The final model excluded land covered by parks and forests. GVI is better suited for use in urban areas because of the scattered distribution of trees and vegetation on roads and in front of buildings and balconies. The third limitation is an inherent feature of GSV. Passing vehicles or other obstacles in certain shooting directions obscure the image, leading to an underestimation of the true greenness value. The fourth limitation is the temporal discontinuity of the greenspace indicators. The temperatures for each month represent the average stable conditions for that month. Our primary focus is green space, which does not change much at the daily level but may vary at the monthly level. As a result, we used monthly temperatures as modeling targets since green space indicators cannot be updated frequently. In this study, we did not consider the human behaviors reacted to the environmental changes. The two-way relationship between human behavior and the environment can be further explored through people's digital footprints [28].

## Conclusion

Green vegetation measured by the GVI in an urban city showed a cooling effect at ambient temperature and reduced the temperature difference. The increase in anthropogenic activities, such as traffic volume and built-up land, has influenced the increase in ambient temperatures. However, the associations were significant at the junction of urban and rural areas in the cities. Street-view-derived ecological indicators are beneficial for understanding urban forms and undermining green vegetation distribution and their association with the urban heat effect at the city scale.

## Supporting information

**S1 Appendix. Average monthly temperature difference in the four seasons from 2018 to 2021 at a cell size of 0.01˚ *0.01˚.**
(DOCX)

**S2 Appendix. Results of the linear mixed model on the average monthly temperature difference.**
(DOCX)

**S3 Appendix. Results of the geographically weighted regression and simple linear regression on average monthly temperature differences.**
(DOCX)

**S4 Appendix. Geographical distributions of the selected significant predictors of monthly average temperature difference.**
(DOCX)

## Author Contributions

**Conceptualization:** Ta-Chien Chan.

**Data curation:** Ta-Chien Chan.

**Formal analysis:** Ping-Hsien Lee, Yu-Ting Lee.

**Funding acquisition:** Ta-Chien Chan.

**Methodology:** Ta-Chien Chan, Jia-Hong Tang.

**Supervision:** Ta-Chien Chan.

**Visualization:** Ping-Hsien Lee, Yu-Ting Lee.

**Writing – original draft:** Ta-Chien Chan.

**Writing – review & editing:** Ta-Chien Chan.

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
