## [Decision Letter · Decision Letter 0]

28 Nov 2023

PONE-D-23-33900Exploring the spatial association between the distribution of temperature and urban morphology with green view indexPLOS ONE

Dear Dr. Chan,

Thank you for submitting your manuscript to PLOS ONE. After careful consideration, we feel that it has merit but does not fully meet PLOS ONE’s publication criteria as it currently stands. Therefore, we invite you to submit a revised version of the manuscript that addresses the points raised during the review process.

We look forward to receiving your revised manuscript.

Kind regards,

Tayyab Ikram Shah, Ph.D.

Academic Editor

PLOS ONE

Journal Requirements:

5. We note that Figure 1 in your submission contain copyrighted images. All PLOS content is published under the Creative Commons Attribution License (CC BY 4.0), which means that the manuscript, images, and Supporting Information files will be freely available online, and any third party is permitted to access, download, copy, distribute, and use these materials in any way, even commercially, with proper attribution. For more information, see our copyright guidelines: http://journals.plos.org/plosone/s/licenses-and-copyright.

A.) You may seek permission from the original copyright holder of Figure 1 to publish the content specifically under the CC BY 4.0 license. 

B.) If you are unable to obtain permission from the original copyright holder to publish these figures under the CC BY 4.0 license or if the copyright holder’s requirements are incompatible with the CC BY 4.0 license, please either i) remove the figure or ii) supply a replacement figure that complies with the CC BY 4.0 license. Please check copyright information on all replacement figures and update the figure caption with source information. If applicable, please specify in the figure caption text when a figure is similar but not identical to the original image and is therefore for illustrative purposes only.

6. We note that Figure 3 - 7 and Appendix 1 in your submission contain map images which may be copyrighted. All PLOS content is published under the Creative Commons Attribution License (CC BY 4.0), which means that the manuscript, images, and Supporting Information files will be freely available online, and any third party is permitted to access, download, copy, distribute, and use these materials in any way, even commercially, with proper attribution. For these reasons, we cannot publish previously copyrighted maps or satellite images created using proprietary data, such as Google software (Google Maps, Street View, and Earth). For more information, see our copyright guidelines: http://journals.plos.org/plosone/s/licenses-and-copyright.

A.) You may seek permission from the original copyright holder of Figure 3 - 7 and Appendix 1 to publish the content specifically under the CC BY 4.0 license.  

B.) If you are unable to obtain permission from the original copyright holder to publish these figures under the CC BY 4.0 license or if the copyright holder’s requirements are incompatible with the CC BY 4.0 license, please either i) remove the figure or ii) supply a replacement figure that complies with the CC BY 4.0 license. Please check copyright information on all replacement figures and update the figure caption with source information. If applicable, please specify in the figure caption text when a figure is similar but not identical to the original image and is therefore for illustrative purposes only.

Reviewers' comments:

Reviewer's Responses to Questions

**Comments to the Author**

1. Is the manuscript technically sound, and do the data support the conclusions?

Reviewer #1: Yes

Reviewer #2: Partly

2. Has the statistical analysis been performed appropriately and rigorously? 

Reviewer #1: Yes

Reviewer #2: Yes

3. Have the authors made all data underlying the findings in their manuscript fully available?

Reviewer #1: Yes

Reviewer #2: Yes

4. Is the manuscript presented in an intelligible fashion and written in standard English?

Reviewer #1: No

Reviewer #2: No

5. Review Comments to the Author

Reviewer #1: The article considers an important research subject. However, the data presented is overloaded information, which needs simplification. There are some questions, which needs to be considered for clarification/explanation of the methods. A few examples are below:

Introduction" Lines 59 - 61: What type of discomfort? Need explanation?

Lines 62-65: This is not clear. Needs rephrasing and further clarification. How the two are indicators? What these indicate to. Instead, both are methods and can be applied for deriving indicators or relationships?

Line 65: Spatial or temporal resolution?

Line 65: Shooting angel limitations: Would factors like shadows of tall buildings and planning be limiting factors?

Line 70: What is meant by thermal comfort?

Line 84: replace "studied" with "recognized" or "admitted" or "considered" or "presented"

Line 89: Which environmental factors?

Line 97: Parameters instead of indicators?

Definition and computation of GVI 2.0

Line 120: How is this different than the first generation. The first generation GVI used six pictures with angles 0 to 180? The difference of splitting a picture to two is not what should be a version difference? Using pixels ultimately is spatial coverage or area, so is not different than using area, which was in the first generation or the referenced modified GVI?

Normalized Difference Vegetation Index (NDVI)

Line 157: What is meant by bird's eye satellite images?

Line 162: With the limiting factors described in the introduction section, was this spatial resolution efficient to capture greenishness in an urban environment? How this could be effective, given high rise building covering a street view from the sky at some angle of view?

Land use data

Line 186: The official classification is 9, 48, and 93. How was this classification made to 09 classes and what was the basis for that?

Altitude and the size of the building area

Line 209: How was this integrated with the other spatial resolution datasets, such as the 250 m MODIS or 30m Landsat? Or it was used independent of those? If used independently, then how was this correlated to the GVI calculated from the GSV?

Results:

Line 264: How was this seasonal variation used? It needs to be explained in the methods and that how this impacted the study/methods?

Line 322: Is it graveyards or different?

Line 323: Graveyards are generally kept greener, a traditional or religious norm across the world. A positive relationship with temperature is considerable.

Reviewer #2: The authors mentioned the effect of small plant using google street view; however, this kind of analysis requires very high-resolution data. It should be mentioned how this data was acquired and preprocessed before feeding it to the deep learning models

The monthly temperature data that the authors has used is very coarse resolution. How will this resolution affect the study, since the authors are talking about ambient temperatures?

In line 61 the authors mentioned that Sateliite data such as NDVI. This is not correct. NDVI and EVI are satellite data derived metrics. The whole statement on line 61 needs to be reevaluated and rewritten

On line 65 what does shooting angle limitations means?

I think the introduction section needs to be rewritten with a proper background of the study and identifying how this is a gap and how it can be addressed using the proposed study

The methods section is okay. Some more details on how the data was preprocessed before giving it to the model should be added.

The discussion needs improvements. The authors can put the results in a proper context and show why this is important and how their results added to the current methods that are being uses.

I would suggest the authors to make the changes and also revise the sentence structure and grammatical issues.

6. PLOS authors have the option to publish the peer review history of their article (what does this mean?). If published, this will include your full peer review and any attached files.

Reviewer #1: No

Reviewer #2: No

---

## [Author Response · Author response to Decision Letter 0]

4 Jan 2024

I have attached the response letter in the system.

---

## [Decision Letter · Decision Letter 1]

7 Mar 2024

PONE-D-23-33900R1Exploring the spatial association between the distribution of temperature and urban morphology with green view indexPLOS ONE

Dear Dr. Chan,

Thank you for submitting your manuscript to PLOS ONE. After careful consideration, we feel that it has merit but does not fully meet PLOS ONE’s publication criteria as it currently stands. Therefore, we invite you to submit a revised version of the manuscript that addresses the points raised during the review process.

We look forward to receiving your revised manuscript.

Kind regards,

Jose Balsa-Barreiro

Academic Editor

PLOS ONE

Journal Requirements:

Additional Editor Comments:

Dear Authors,

Following the review process and the successful addressing of the reviewers' concerns, I am pleased to recommend this work for publication. From my perspective, I would like to suggest some references to current works that precisely explore a reciprocal relationship between urban morphology and human behavior. Among them is the work edited by MIT Press on Digital Ethology - (Strüngmann Forum Reports) by Tomas Paus & Hye-Chung Kum, which offers an edited collection that delves deeply into how humans transform their environments and how these environments, in turn, shape humans. Countless permutations of physical, built, and social environments surround us in space and time, influencing the air we breathe, our comfort levels, our daily activity patterns, and our social interactions. Additionally, I recommend the chapter "How cities influence social behavior," which highlights not only urban morphology but also physical aspects such as temperature and green spaces.

Best regards,

Reviewers' comments:

Reviewer's Responses to Questions

**Comments to the Author**

1. If the authors have adequately addressed your comments raised in a previous round of review and you feel that this manuscript is now acceptable for publication, you may indicate that here to bypass the “Comments to the Author” section, enter your conflict of interest statement in the “Confidential to Editor” section, and submit your "Accept" recommendation.

Reviewer #1: All comments have been addressed

Reviewer #2: All comments have been addressed

2. Is the manuscript technically sound, and do the data support the conclusions?

Reviewer #1: Yes

Reviewer #2: Yes

3. Has the statistical analysis been performed appropriately and rigorously? 

Reviewer #1: I Don't Know

Reviewer #2: N/A

4. Have the authors made all data underlying the findings in their manuscript fully available?

Reviewer #1: No

Reviewer #2: Yes

5. Is the manuscript presented in an intelligible fashion and written in standard English?

Reviewer #1: Yes

Reviewer #2: Yes

6. Review Comments to the Author

Reviewer #1: Thank you for addressing the comments and making changes to the manuscript accordingly. The manuscript is improved and the reviewer has no further comments.

Reviewer #2: (No Response)

7. PLOS authors have the option to publish the peer review history of their article (what does this mean?). If published, this will include your full peer review and any attached files.

Reviewer #1: No

Reviewer #2: No

---

## [Author Response · Author response to Decision Letter 1]

7 Mar 2024

Dear Editor,

We have revised and submitted our manuscript, “Exploring the spatial association between the distribution of temperature and urban morphology with green view index,” to be considered for publication in PLoS One. We have taken editor’s suggestion to add the citation in our limitation. However, we don't see any further information about the book, which will be available this summer from the Amazon.com. So it's really hard for us to discuss more about it.

“In this study, we did not consider the human behaviors reacted to the environmental changes. The two-way relationship between human behavior and the environment can be further explored through people’s digital footprints [28].”

We prepared both a version of the manuscript with tracked changes and a clean version for your reference. We hope that the editor will be satisfied with our response.

Yours sincerely,

Ta-Chien Chan, Ph.D.

Research Fellow

Research Center for Humanities and Social Sciences, Academia Sinica, Taiwan

---

## [Editor Report · Decision Letter 2]

25 Mar 2024

Exploring the spatial association between the distribution of temperature and urban morphology with green view index

PONE-D-23-33900R2

Dear Dr. Chan,

We’re pleased to inform you that your manuscript has been judged scientifically suitable for publication and will be formally accepted for publication once it meets all outstanding technical requirements.

Kind regards,

Jose Balsa-Barreiro

Academic Editor

PLOS ONE

Additional Editor Comments (optional):

Dear Authors,

After conducting a thorough review of your manuscript and considering the satisfactory addressing of my concerns, I am pleased to recommend the acceptance of your present manuscript for publication

Sincerely,

The Academic Editor
---

## [Editor Report · Acceptance letter]

26 Apr 2024

PONE-D-23-33900R2 

PLOS ONE

Dear Dr. Chan, 

I'm pleased to inform you that your manuscript has been deemed suitable for publication in PLOS ONE. Congratulations! Your manuscript is now being handed over to our production team.

Kind regards, 

on behalf of

Dr. Jose Balsa-Barreiro 

Academic Editor

PLOS ONE